# Learning Decentralized Multi-Biped Control for Payload Transport

**Bikram Pandit,    Ashutosh Gupta,    Mohitvishnu S. Gadde,    Addison Johnson,**
**Aayam Kumar Shrestha,    Helei Duan,    Jeremy Dao,    Alan Fern**

Collaborative Robotics and Intelligent Systems (CoRIS) Institute, Oregon State University
{panditb, guptaash, gaddem, johnsadd,
shrestaa, duanh, daoje, afern}@oregonstate.edu

**Project webpage: decmbc.github.io**

**Abstract:** Payload transport over flat terrain via multi-wheel robot carriers is well-understood, highly effective, and configurable. In this paper, our goal is to provide similar effectiveness and configurability for transport over rough terrain that is more suitable for legs rather than wheels. For this purpose, we consider multi-biped robot carriers, where wheels are replaced by multiple bipedal robots attached to the carrier. Our main contribution is to design a decentralized controller for such systems that can be effectively applied to varying numbers and configurations of rigidly attached bipedal robots without retraining. We present a reinforcement learning approach for training the controller in simulation that supports transfer to the real world. Our experiments in simulation provide quantitative metrics showing the effectiveness of the approach over a wide variety of simulated transport scenarios. In addition, we demonstrate the controller in the real-world for systems composed of two and three Cassie robots. To our knowledge, this is the first example of a scalable multi-biped payload transport system.

**Keywords:** Multi-robot Transport, Bipedal locomotion, Reinforcement Learning

## 1   Introduction

Multi-wheel carriers are the most common way to transport payloads across well-structured terrain due to their ease of control, effectiveness, and configurable wheel arrangements [1]. Further, researchers have also considered wheeled transport that can be reconfigured on the fly via multiple co-operating wheeled robots [2]. While these transport systems excel in structured environments, they struggle in unstructured and challenging terrain, which limits their general applicability. This work aims to overcome such limitations while retaining the control and reconfigurability of multi-wheel systems. To achieve this, we propose the *multi-biped transport* problem, where wheels or wheeled robots are replaced with bipedal robots that can be arbitrarily attached to carriers. This setup allows for transport over various terrains, potentially extending to more challenging environments [3, 4, 5].

Multi-biped transport presents significant challenges. Legged robots are generally unstable and highly dynamic, making them much harder to control compared to wheeled robots. Maintaining stability while transporting payloads and coordinating multiple legged-robots adds further complexity to the control problem [6]. However, the use of multiple bipeds offers distinct advantages over alternatives such as large quadrupeds, providing greater flexibility in payload transport. This approach allows the number and arrangement of robots to be tailored based on the weight and size of the payload. Such versatility opens up potential applications in various areas including construction, moving systems, warehouses, agriculture, and delivery systems.

Prior works on multi-legged transport systems have primarily focused on model-based approaches that rely on accurate modeling of robot interaction and explicit inter-robot communication, which limit their reconfigurability and practicality [7, 8]. While learning-based approaches have emerged as promising alternatives [9, 10, 11, 12, 13, 14, 15, 16, 17, 18] for various control problems, existing methods on multi-robot system often assume centralized control with strong assumptions on system observability, leading to limited scalability, robustness, and vulnerability to point of failure [19].

8th Conference on Robot Learning (CoRL 2024), Munich, Germany.

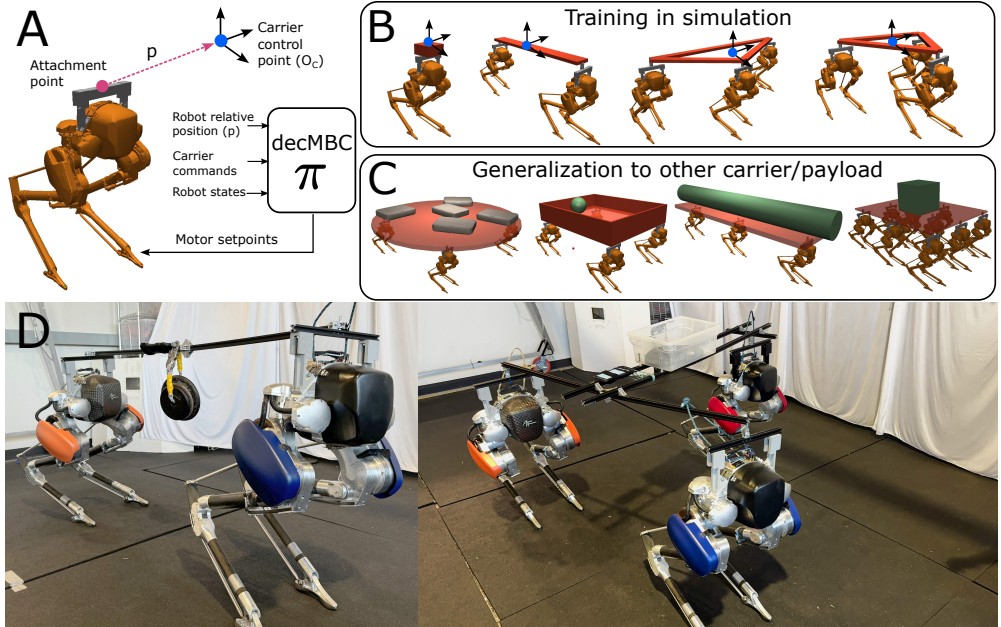

Figure 1: **A)** Illustrates the Decentralized Multi-Bipedal Controller (decMBC), which is a single controller deployable to any number of robots for carrier transport without requiring any inter-robot communication. **B)** Shows the distribution of multi-robot configurations used in the training curriculum for decMBC in simulation. **C)** Highlights unseen carrier and payload scenarios which the decMBC policy was evaluated for in the simulation. **D)** Showcases sim-to-real deployment where each robot running a copy decMBC policy on hardware.

To overcome these challenges, we propose an architecture and training approach for a decentralized Multi-Biped Controller (decMBC). This controller supports high-level motion control of a carrier that is attached to any arbitrary configuration of bipedal robots. Importantly, to maintain maximum reconfigurability, the decMBC is decentralized in the sense that each biped can be controlled using only information about its local state and its relative position with respect to a desired reference point on the carrier. In addition, the decMBC can be applied, without retraining, to any number of bipeds that are attached to a carrier in arbitrary stable configurations. The proposed decMBC is trained using decentralized reinforcement learning (RL) over a distribution of robot-carrier configurations carefully selected to provide generalization to arbitrary biped configurations and facilitate sim-to-real transfer. The approach is easy to implement and can be used with a wide range of existing single-agent control architectures and RL algorithms, making it highly adaptable and accessible.

We validate the effectiveness and flexibility of our approach through extensive experiments across varying numbers of robot and carrier configurations. Our quantitative results in simulation highlight the system's ability to maintain stability, coordinate effectively, and adapt to different payload transport tasks. Furthermore, we demonstrate sim-to-real transfer to multi-biped systems involving two and three Cassie robots. To the best of our knowledge, this is the first multi-leg transport system to demonstrate this level of reconfigurability in simulation and the real world.

## 2 Related Work

Multi-robot transport has attracted significant research attention due to its potential to overcome the limitations of single-robot systems, especially in handling large, heavy payloads and navigating diverse real-world environments. Several works have focused on wheeled robots for collaborative transport which developed a composite connector for non-holonomic mobile robots, but their work primarily focused on well-structured environments [2]. Moreover, Eoh and Park [20] propose a cooperative object transport controller using deep reinforcement learning which uses simple non-holonomic wheeled robots to learn the collaborative pushing of an object to its goal position. While their approach shows promise, it assumes that the robots can sense the positions of the object and the goal with respect to each robot. This makes scaling the approach for arbitrary robot-carrier configurations difficult.

Similarly, other works have considered predictive control and decentralized approaches. Fawcett et al. [21] proposed a predictive control approach for robust locomotion of three holonomic constraint quadrupeds, but it did not allow for formation changes or scaling to more agents. Zhang et al. [22] introduced a decentralized control approach using DQN controllers, but they strongly assumed that each robot could perfectly sense the state of the payload and every robot. Lin et al. [23] develop an end-to-end decentralized deep reinforcement learning policy to learn multi-robot control directly from LiDAR measurements. They employ centralized learning and decentralized execution to deploy on holonomic UGVs. However, their approach relies on the assumption that the team-level action should be a concatenation of all individual actions, which limits its flexibility.

On the other hand, researchers have also explored multi-legged transport systems. Kim et al. [7, 8] proposed layered control strategies and control barrier functions for holonomically connected quadrupeds, but their approaches were not generalized for any number of robots and relied on strong assumptions. Ji et al. [19] demonstrated a reinforcement learning strategy for four quadrupedal robots executing a collaborative task, but their approach was limited to a fixed configuration and number of agents, and the results are shown only in simulation. Other notable works include Horyna et al. [24], who developed a collaborative transport control for two multi-copters using a master-slave configuration, and Pei et al. [25], who demonstrated a multi-robot system for collaboratively transporting objects in a net spanning across all agents. These approaches faced challenges in scaling with the number of agents and had restrictions on the size of the net and topology of team formation.

In this work, we focus on the most primitive legged base i.e., bipedal robots. For bipedal robots, Dao et al. [26] investigate the potential of reinforcement learning and sim-to-real transfer for bipedal locomotion under dynamic loads, relying solely on proprioceptive feedback. While they demonstrated effectiveness across a wide range of external forces, their approach faced limitations. Specifically, their method was designed only for a single robot and was constrained by weight, making scalability to multiple robots or handling heavier loads challenging.

Building upon these prior works, our proposed decentralized Multi-Biped Controller (decMBC) aims to overcome the limitations of existing approaches by enabling decentralized control and generalization to arbitrary carrier configurations of legged robots. Unlike the works that focus on wheeled robots, our decMBC has the potential to handle the complexity and dynamics of legged robots in challenging terrain. Furthermore, our decentralized architecture enhances the scalability and robustness of the system, making it more practical for real-world deployment.

## 3 Problem Formulation

We consider multi-biped transport problems where a set of $N$ identical bipedal robots are rigidly attached to the base of a payload carrier. We model the *carrier base* $B \subset \mathbb{R}^2$ as a rigidly connected planar surface with a finite area. Each carrier base is assumed to contain the origin, which is referred to as the *carrier control point* $O_C$ [see Figure 1 (A)]. An $N$ robot *attachment configuration* for a base $B$ is a set of points $P = \{p_1, \ldots, p_N\}$, such that $p_r \in B$, which indicates where each robot is attached. In this work, we assume ball joint attachments where each robot has free movements in all three rotational directions relative to the attachment but constrains movements in linear directions. However, our approach is easily adapted to other types of joints and non-planar rigid platform bases.

Our goal is to control the height and linear and angular velocity of the carrier control point via the combined motor actions of the robots. To support flexible applications, we consider a decentralized controller architecture that avoids inter-robot communication and is flexible to varying numbers of robots $N$ and configurations $P = \{p_1, \ldots, p_N\}$. The *decentralized Multi-Biped Controller (decMBC)* is defined by a single-biped controller $\pi$. At each time-step, $t$, this controller is independently applied to each robot $r \in \{1, \ldots, N\}$ to compute robot actions $a_t^r = \pi(o_t^r, c_t)$, where $o_t^r$ is the local observation and $c_t$ is the carrier command.

In our work, the command $c_t \in \mathbb{R}^4$ specifies the target linear xy velocity, angular yaw velocity, and height of the control point. The local observation $o_t^r$ contains: 1) the local *proprioceptive state* $s_t^r$ of the robot, including the joints/motor positions and velocities, base orientation, and angular velocity, 2) the *attachment position* $p_r$ of the robot relative to the control point, and 3) the *relative*

*yaw orientation* $\theta_t^r \in \mathbb{R}$ of the robot base with respect to the control point orientation axis. Each of our actions $a_t^r$ specifies the target PD motor set points of robot $r$, which includes 10 motors (5 in each leg) for the Cassie biped used in our experiments. Note that $\pi$ can be applied to any number of robots in any configuration provided that each robot can obtain its relative position $p_r$ and orientation $\theta_t^r$. The challenge is to learn such a decentralized control structure that is effective and robust.

# 4 Learning Approach

We formulate learning the decMBC as a reinforcement learning (RL) problem, as it involves solving a complex physical control task without supervision for low-level actions. Our approach employs Independent Proximal Policy Optimization (IPPO) [27], where the shared decMBC controller $\pi$ is trained using data collected from multiple robots operating in parallel, without inter-robot communication. Below we describe the architecture of decMBC, training environment and curriculum, reward function, and details of IPPO training procedure.

## 4.1 Neural Network Architecture

Following prior work on bipedal locomotion [28], we represent the decMBC using a recurrent neural network architecture, which allows for history-dependent action selection via the recurrent internal memory. Specifically, our network consists of two Long Short-Term Memory (LSTM) [29] layers, each containing 64-dimensional hidden states. These layers process the decMBC input at each time step to form the internal hidden state. This state is then processed by a linear layer that produces a 10-dimensional action vector, specifying the 5 motor set points for each of Cassie's legs.

## 4.2 Episode Generation and Curriculum

Our IPPO training approach uses training data generated by executing training episodes involving one or more robots. During each episode, transitions from each robot are aggregated into a single replay buffer that is used to optimize a single actor and critic using the PPO objective [14]. Details of the PPO hyperparameters are in the Appendix E. The single actor at the end of training is taken to be the decMBC controller. Below, we describe the episode generation process, training configurations, and curriculum used to train the MBC.

**Episode Generation.** At the start of each training episode, we generate a random robot configuration according to the current curriculum stage (see below), including randomized weights on the rigid bars connecting robots. During an episode, the decMBC is used to control the $N$ robots, given a randomized sequence of carrier commands. Each command is executed for a random number of time steps in the interval [100,450]. In addition, depending on the curriculum stage, random perturbation forces (0 to 50 N) are applied to the robot's pelvis and the carrier. Each episode ends after 500 time-steps (10 seconds) or a termination condition that checks whether a robot or the carrier has fallen, which is detailed in the Appendix C.

**Training Configurations.** Our current simulation time scales super-linearly as the number of robots grows. For this reason, we limit the number of robots used during training to $N \leq 3$. Empirically, we have found that using training configurations with $N \leq 3$ leads to strong performance for $N > 3$. We generate random configurations for $N \in \{1, 2, 3\}$ as follows: For $N = 1$, a single Cassie robot is used with a small carrier, where the attachment point is located at the center of the carrier. For $N = 2$, two Cassie robots are used with their attachment points being endpoints of a rigid bar. The carrier control point is located anywhere within a meter away from either of the robot's positions. For $N = 3$, an arbitrary triangular frame is formed by connecting three rigid bars. Here, the attachment points for three Cassie robots are the vertices of the triangular frame. In this configuration, the carrier control point can be randomized anywhere enclosed in the triangular frame [Figure 1 (B)]. Further details are given in Appendix A.

**Training Curriculum.** We initialize the decMBC controller and critic randomly and train it via IPPO using a 4 stage curriculum, where each stage is defined by the configurations and disturbances considered. *Stage 1* involves training with one robot ($N = 1$) with the objective of learning robust locomotion gaits operating for carrier commands at its attachment point. *Stage 2* involves training with the addition of perturbation force and torsion force on the carrier such that a robot is resistant to the disturbance forces. *Stage 3* involves training with the addition of two-robot and three-robot configurations without any perturbation and torsion forces. This stage ensures each robot learns to

operate on the carrier control point randomly placed away from its attachment point. *Stage 4* involves applying perturbation and torsion forces on the robot's base and carrier in addition to randomizing the mass of the rigid bar connecting the robots. This ensures each robot experiences disturbance forces and varying masses while operating with respect to the carrier control point.

## 4.3 Reward Design

Our reward function is designed to produce stable locomotion behavior while ensuring the robots collaboratively follow the commanded motion of the carrier control point. For each robot $i$ in N-robot configuration, where $i \in \{1, \ldots, N\}$, the reward $r_i$ is composed of two components: the local robot reward $r_i^L$ and global reward $r^G$, where $r_i = r_i^L + r^G$ at each timestep throughout the episode. The local reward $r_i^L$ is designed to produce stable locomotion behavior of each robot, including maintaining consistent base height, penalizing jerky motion, regularizing proper foot alignment and the stepping frequency, with single-contact reward as proposed by van Marum et al. [30]. Furthermore, the local reward minimizes torque of the motors to penalize excessive energy expenditure and minimizes the force exerted at the attachment point at the carrier during standing, ensuring that robots do not pull or push against each other but hold the carrier steady. The global reward $r^G$ focuses on the performance of the combined action of all robots in following carrier commands. It aims to minimize the deviation between the commanded and actual x-velocity, y-velocity, angular yaw velocity, and height of the carrier control point. We detail the reward components in Appendix B.

## 5 Experiments and Results

We conduct experiments to empirically evaluate the decMBC on the Cassie bipedal robot platform. The primary objective is to assess the effectiveness and adaptability to coordinate an arbitrary number of robots, maintain stability, and adapt to diverse transport scenarios. We also discuss the scalability and generalizability of our approach across a range of practical settings including real-world transfer to multi-biped systems with two and three Cassie robots.

**Experimental Procedure and Metrics.** We compute metrics such as drift, failure rate, and power consumption across various configurations to evaluate our decMBC policy. We also measure these metrics to compare our approach against other approaches. For each of these metrics across various carrier configurations, we run 1000 episodes for a maximum of 20 seconds, each with a fixed carrier command such as hold still, move forward, move sideways, and turn in-place. *Drift* measures the displacement of the carrier in the x/y direction and its orientational drift from its expected orientation and considers episodes that last for at least 1 second. We do not apply any perturbation forces for measuring drift. A negative and positive sign in drift indicates that the carrier is behind or ahead of the expected global position for the desired carrier command. *Failure rate* is calculated as the percentage of episodes that terminate before reaching 20 seconds. This is evaluated under varying perturbation forces and payload masses fixed on the top of the carrier (except dynamic payload on Table 2). We apply ranges of perturbation forces and change payload masses in the carrier to measure the failure rate. Additionally, we calculate the *Power consumption* by all robots combined for each configuration.

### 5.1 Evaluating Scalability and Reconfigurability

**Varying Numbers of Robots:** We first evaluate the performance of the decMBC for varying numbers of robots. For this purpose, we vary the number of robots from 2 to 10 using the same solid rectangular carrier. The robots are placed in a manner that ensures they are evenly distributed to maintain static stability. This typically involves positioning robots at the corners and, when necessary, placing additional robots in a central position to maintain balance (see Figure 2). In order to provide a reference point for strong performance, we evaluate the metrics for a single robot policy (1-R*) trained with its floating base as the control point. We consider achieving multi-biped performance that is comparable to 1-R* as a practical gauge of success.

Table 1 gives results for the reference 1-R* and decMBC policies across the various control metrics. We first see that there is minimal performance variation across the range of 2 to 10 robots, showing that the decMBC is generalizing beyond its training scenarios of 1 to 3 robots. One exception is for $N = 2$ which performs significantly worse on the turn in-place test, though still achieves relatively

small errors on other commands. This could be because of compromised stability arising from limited lateral support provided by two robots for a rectangular carrier. However, for $N > 2$, the maximum orientation drift observed is $-7.47°$, which is relatively small given the commanded $300°$ rotation. Overall, we see that the multi-robot performances are generally comparable to 1-R*.

Figure 3 presents failure rates for increasing carrier perturbation forces and payload masses as the number of robots varies, along with the power usage. The results show a sharp decrease in failure rate as the number of robots increases, achieving nearly zero failures. The cost for the robustness is increased power consumption. Notably, the power consumption scales approximately linearly with the number of robots, suggesting that the robots cooperate effectively without exerting excessive counter-forces on each other.

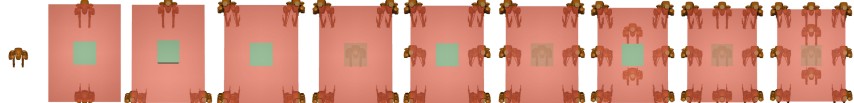

Figure 2: Configuration across varying numbers of robots for a rectangular carrier with payload placed at the center. From left to right, we have 1-R*, and number of robots ranging from 2 to 10.

| Number of robots | Hold still $V_x=0, V_y=0, \omega=0$ | | | Move forward $V_x=1m/s, V_y=0, \omega=0$ | | | Move sideways $V_x=0, V_y=0.25m/s, \omega=0$ | | | Turn in-place $V_x=0, V_y=0, \omega=15°/s$ | | |
|---|---|---|---|---|---|---|---|---|---|---|---|---|
| | $\Delta x$ | $\Delta y$ | $\Delta \theta$ | $\Delta x$ | $\Delta y$ | $\Delta \theta$ | $\Delta x$ | $\Delta y$ | $\Delta \theta$ | $\Delta x$ | $\Delta y$ | $\Delta \theta$ |
| 1-R* | 0.07 | 0.00 | -0.51 | -1.48 | 0.37 | 0.54 | 0.24 | -0.56 | 1.37 | -0.09 | 0.03 | -1.85 |
| 2 | 0.16 | 0.28 | 1.29 | -1.64 | 0.93 | 0.91 | 0.03 | -0.21 | 0.44 | 0.19 | 0.12 | -12.98 |
| 3 | 0.06 | 0.21 | 0.26 | -1.54 | 0.78 | -0.55 | 0.13 | -0.70 | -0.46 | 0.16 | 0.21 | -6.00 |
| 4 | 0.06 | 0.08 | -1.06 | -1.54 | 0.91 | -0.51 | 0.20 | -0.32 | 0.40 | 0.08 | 0.38 | -6.71 |
| 5 | 0.08 | 0.13 | -1.29 | -1.56 | 0.72 | -0.44 | 0.31 | -0.54 | 0.57 | -0.03 | 0.11 | -7.04 |
| 6 | 0.05 | 0.07 | -0.79 | -1.60 | 0.77 | -0.42 | 0.27 | -0.42 | 0.46 | 0.04 | 0.29 | -6.88 |
| 7 | 0.06 | 0.10 | -0.78 | -1.63 | 0.64 | -0.50 | 0.35 | -0.59 | 0.51 | -0.05 | 0.12 | -7.47 |
| 8 | 0.03 | 0.10 | -0.80 | -1.64 | 0.71 | -0.38 | 0.27 | -0.58 | 0.58 | 0.05 | 0.22 | -5.99 |
| 9 | 0.04 | 0.11 | -0.65 | -1.59 | 0.74 | -0.42 | 0.27 | -0.63 | 0.93 | 0.02 | 0.12 | -6.76 |
| 10 | 0.05 | 0.13 | -0.54 | -1.63 | 0.72 | -0.15 | 0.32 | -0.67 | 0.71 | 0.02 | 0.10 | -6.28 |

Table 1: Positional ($\Delta$x-y in m) & orientational ($\Delta\theta$ in °) drift observed after 20 seconds for each command with no perturbation force and payload mass of 20kg.

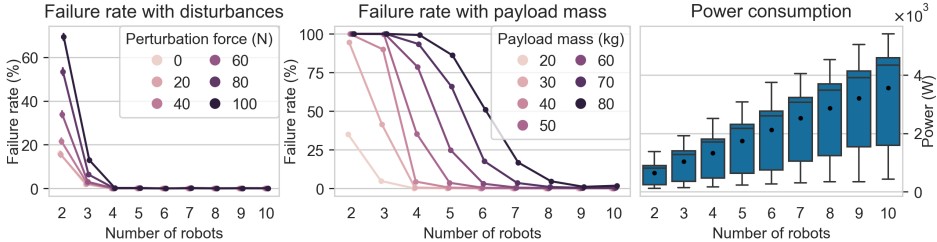

Figure 3: (Left) Failure rate for varying number of robots carrying a payload mass of 20kg experiencing various perturbation forces from 0 to 100N. (Center) Failure rate for varying numbers of robots for different payload masses from 20 to 80kg averaged over all perturbation forces. (Right) Power consumed per second by all robots combined and averaged over all perturbation forces and payload masses.

**Varying carrier configuration and payload:** In real-world scenarios, payload transport tasks often involve diverse carrier shapes and sizes to accommodate various objects and environments. We demonstrate the adaptability of our approach to such practical settings by evaluating the decMBC's performance on three distinct carrier configurations [see Figure 1 (C)], each representing a different real-world use case. The first configuration, *Sacks*, involves four robots attached to a circular carrier with five sacks of varying weights, totaling 35 kg, randomly fixed on top. The second configuration, *Log*, uses a long plank as a carrier, with three robots attached collinearly, carrying a cylinder that is fixed on the carrier, simulating a log weighing 20 kg. The final configuration, *Dynamic*, shows four robots attached to a carrier that is a container holding a rolling ball with a total mass of 30 kg.

We observe from Table 2 that the controller maintains near-zero drift across all commands and configurations, with the exception of orientation drift during the turn in-place command for dynamic loads. In this case, the drift is $-77.95°$, representing an approximately 25% error from the commanded $300°$. This discrepancy may be attributed to the unpredictable motion of the rolling ball, which alters the momentum of the carrier. Nevertheless, given that the decMBC policy has never seen these types of payloads during training, the failure rate seems reasonably low.

| | Hold still $V_x$=0, $V_y$=0, $\omega$=0 | | | Move forward $V_x$=1m/s, $V_y$=0, $\omega$=0 | | | Move sideways $V_x$=0, $V_y$=0.25m/s, $\omega$=0 | | | Turn in-place $V_x$=0, $V_y$=0, $\omega$=15°/s | | | Mean Failure Rate (%) |
|---|---|---|---|---|---|---|---|---|---|---|---|---|---|
| | $\Delta x$ | $\Delta y$ | $\Delta \theta$ | $\Delta x$ | $\Delta y$ | $\Delta \theta$ | $\Delta x$ | $\Delta y$ | $\Delta \theta$ | $\Delta x$ | $\Delta y$ | $\Delta \theta$ | |
| Sacks | 0.11 | 0.09 | -1.60 | -1.49 | 1.30 | -0.60 | -0.07 | 0.54 | 0.96 | 0.06 | 0.55 | -13.15 | 14.71 |
| Log | 0.23 | 0.32 | 1.12 | -1.14 | 0.50 | 0.93 | 0.49 | -0.70 | 1.36 | -0.02 | 0.05 | -7.86 | 8.58 |
| Dynamic | 0.15 | 0.06 | -0.57 | -3.86 | 0.44 | -0.25 | 0.07 | -0.54 | -0.24 | -0.07 | -0.20 | -77.95 | 17.90 |

Table 2: Positional ($\Delta$x-y in m), orientational ($\Delta\theta$ in °) drift with no perturbation forces observed after 20 seconds & mean failure (%) averaged over all commands, and perturbation forces.

## 5.2 Comparison with Centralized and Specialized policies

In this section, we compare our decentralized and adaptable decMBC with two types of straightforward but inflexible systems: centralized policies and specialized policies. The aim is to demonstrate that our approach, despite its flexibility and generalizability, does not significantly compromise performance compared to these more rigid alternatives. A *centralized policy* is one where all robots' states are used to produce actions for each robot from a single controller. This approach relies on a comprehensive view of the entire system to make coordinated decisions. A *specialized policy*, on the other hand, is designed for a specific carrier configuration with a fixed number of robots. It is trained to perform optimally under these specific conditions but lacks the flexibility to adapt to different setups.

From our observations in Table 3 and Figure 4, the drift values for our approach are reasonably close to the centralized policy. Interestingly, our approach outperforms the centralized policy for two-robot configuration on failure rate, but the performance worsens on the three-robot configuration. We can also observe that the power consumption of our approach is slightly higher than that of the centralized approach. This difference could be attributed to the centralized controller's ability to produce better coordination due to its full observation of the system state.

Interestingly, from the observations in Table 4 and Figure 5, our decentralized approach outperforms the specialized policy for *Rectangle* and *T-shape* configurations in terms of failure rate, with a cost of slightly more power consumption. Our argument for this result is that the decentralized policy, through its adaptive nature, may have generalized to alternative variations of similar configurations, making it more robust to unexpected disturbances.

While our decentralized approach shows comparable or better performance in many scenarios, centralized policies theoretically have higher potential capacity due to their access to global information. However, our attempts at training centralized controllers have not yielded substantial improvements, possibly because our decentralized architecture is sufficiently rich for near-optimal performance under rigid attachments, or because the higher dimensionality of the centralized approach makes it more challenging to train effectively. For instance, training a centralized controller for a 3-robot configuration took about 1000 hours to reach peak reward, compared to 245 hours for our decentralized policy. This suggests that while centralized policies may have higher theoretical capacity, our decentralized approach offers a favorable trade-off between performance and training efficiency.

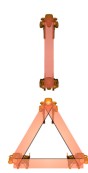

| | | Hold still $V_x$=0, $V_y$=0, $\omega$=0 | | | Move forward $V_x$=1m/s, $V_y$=0, $\omega$=0 | | | Move sideways $V_x$=0, $V_y$=0.25m/s, $\omega$=0 | | | Turn in-place $V_x$=0, $V_y$=0, $\omega$=15°/s | | |
|---|---|---|---|---|---|---|---|---|---|---|---|---|---|
| | Policy $\pi$ | $\Delta x$ | $\Delta y$ | $\Delta \theta$ | $\Delta x$ | $\Delta y$ | $\Delta \theta$ | $\Delta x$ | $\Delta y$ | $\Delta \theta$ | $\Delta x$ | $\Delta y$ | $\Delta \theta$ |
| 2 | Centralized | 0.35 | -0.29 | 0.55 | -1.68 | 0.43 | 0.93 | 0.44 | -1.45 | 0.24 | -0.18 | 0.05 | -3.74 |
| 2 | Ours | 0.07 | 0.18 | 1.15 | -1.03 | 0.97 | 1.00 | 0.71 | -0.73 | -1.14 | -0.20 | -0.07 | -4.78 |
| 3 | Centralized | 0.02 | 0.06 | -0.09 | -1.53 | 0.08 | 0.85 | 0.19 | -0.68 | 0.20 | -0.09 | -0.16 | -1.98 |
| 3 | Ours | 0.17 | 0.17 | 0.33 | -1.66 | 0.57 | -0.14 | -0.01 | -0.44 | -1.01 | 0.08 | -0.07 | -6.38 |

Table 3: Positional ($\Delta$x-y in m) & orientational ($\Delta\theta$ in °) drift observed after 20 seconds for each command with no perturbation force, comparing Centralized and Our approach.

## 5.3 Sim-to-real Transfer

We evaluated our policy on real hardware consisting of two and three bipedal robots, Cassie, attached at the endpoints of an I-shaped and T-shaped carrier made from aluminum extrusion with a weight holder. We also added a payload on the weight holder and used it as a carrier control point [Figure 1 (D)]. We deployed the decMBC policy on each of the robots, running independently, providing

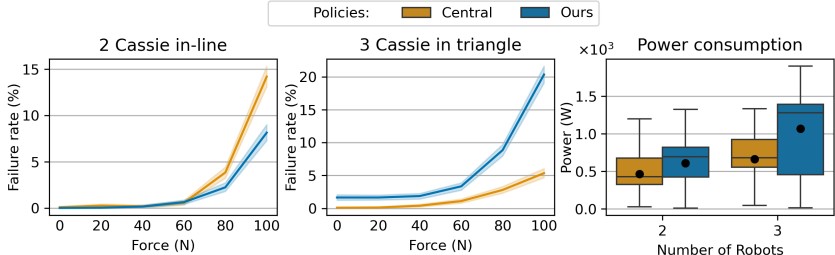

Figure 4: (Left, Center) Failure rate for 2-Cassie in-line and 3-Cassie in triangular configuration experiencing various perturbation forces from 0 to 100N. (Right) Power consumed per second by all robots combined and averaged over all perturbation forces and commands for Centralized and Our policy.

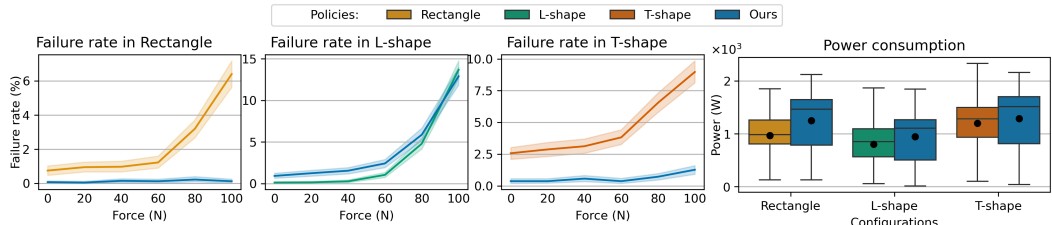

| | | | Hold still $V_x=0, V_y=0, \omega=0$ | | | Move forward $V_x=1m/s, V_y=0, \omega=0$ | | | Move sideways $V_x=0, V_y=0.25m/s, \omega=0$ | | | Turn in-place $V_x=0, V_y=0, \omega=15°/s$ | | |
|---|---|---|---|---|---|---|---|---|---|---|---|---|---|---|
| | Policy $\pi$ | $\Delta x$ | $\Delta y$ | $\Delta \theta$ | $\Delta x$ | $\Delta y$ | $\Delta \theta$ | $\Delta x$ | $\Delta y$ | $\Delta \theta$ | $\Delta x$ | $\Delta y$ | $\Delta \theta$ |
| Rectangle | Specialized | 1.29 | 0.40 | 1.34 | -1.15 | -1.56 | 1.39 | 1.08 | 0.12 | 1.22 | -0.06 | 0.01 | -3.89 |
| | Ours | 0.06 | 0.08 | -1.06 | -1.54 | 0.91 | -0.51 | 0.20 | -0.32 | 0.40 | 0.08 | 0.38 | -6.71 |
| L-shape | Specialized | -0.69 | 0.31 | -1.33 | -2.18 | 0.37 | -3.23 | -0.22 | -0.69 | -1.72 | 0.10 | 0.09 | -10.39 |
| | Ours | 0.12 | -0.04 | 0.37 | -0.95 | 0.63 | -0.45 | 0.43 | -1.22 | 2.46 | -0.85 | -0.59 | -18.65 |
| T-shape | Specialized | 1.37 | 0.57 | -3.89 | -1.38 | -0.42 | -2.97 | 0.33 | -0.38 | -1.05 | -0.32 | -0.11 | -20.28 |
| | Ours | 0.23 | 0.03 | -0.60 | -1.00 | 0.74 | -0.28 | 0.60 | -1.41 | 2.41 | -0.61 | -0.58 | -16.23 |

Table 4: Positional ($\Delta$x-y in m) & orientational ($\Delta\theta$ in °) drift observed after 20 seconds for each command with no perturbation force, comparing Specialized and Our approach.

Figure 5: (First, Second, Third) Failure rate for three configurations experiencing various perturbation forces from 0 to 100N. (Fourth) Power consumed per second by all robots combined averaged over all perturbation forces and commands for three configurations.

their relative positions with respect to the carrier control point. We use a joystick to broadcast the same carrier command values to all the robots using the UDP protocol. We commanded x-velocity, y-velocity, angular yaw velocity, and height to maneuver the payload from one point to another. We observed that it was successfully able to maneuver different payloads according to various user commands for an extended period of operation.

## 6 Limitations

Despite the promising results, there are clear limitations to address in future work. First, the decMBC is not explored on diverse terrain types beyond flat surfaces, eliminating the understanding of its performance in more challenging environments. Second, real-world experiments are demonstrated to up to three Cassie robots, despite the simulation results showing the scalability of up to 10 robots. Third, the controller relies solely on proprioceptive sensing and relative robot positions without incorporating additional sensing modalities like camera perception which could enhance its capabilities in complex environments. Fourth, there are several practical challenges associated with multi-biped systems that require further consideration, including higher costs, increased points of failure, perception difficulties due to robot proximity, and the complexity of deployment and maintenance. These factors highlight important trade-offs between system flexibility and operational practicality. Finally, although our policy is trained with a wide range of relative positions of each robot with respect to the carrier point, which should cover most real-world scenarios, our policy does not guarantee it may work for positions beyond the range we have set during training.

**Acknowledgments**

This material is based on work supported by the National Science Foundation under Grants 2321851 and IIS-1724360 and USDA 2021-67021-35344. We thank all the reviewers for their helpful feedback, which improved the quality of this paper. In addition to the authors, we also appreciate the support from our colleagues, Pranay Dugar and Bart van Marum, for their assistance with robot operation and repair.

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

# Appendix

## A    Implementation details

In this section, we describe the implementation details of our decMBC approach, focusing on the configuration generation, and randomization of parameters used during training.

For generating $N$-robot configuration during training, where $N \in \{1, 2, 3\}$, we choose relative position $p_i \ \forall i \in \{1 \dots N\}$ with respect to the carrier control point which is a 2D polar coordinate indicated by $(R_i, \Theta_i)$. Figure 6 shows an example of robot configurations during training. In Figure 6a ($N = 1$), the carrier control point is located at the base of the robot, therefore relative position $(R_1, \Theta_1)$ is always zero. In Figure 6b ($N = 2$), we randomize carrier control point location up to 1 meter from any of the robots, and similarly, in Figure 6c ($N = 3$), we randomize carrier control point to be located anywhere enclosed within the triangular frame, which produces a range of relative position $(R_i, \Theta_i)$. Furthermore, in each of these configurations, we randomize the mass of the rigid bar connection as shown in Table 5.

The ranges of relative positions and masses seen during training generalize for various carrier configurations with an arbitrary number of robots and payloads as observed in the experiments.

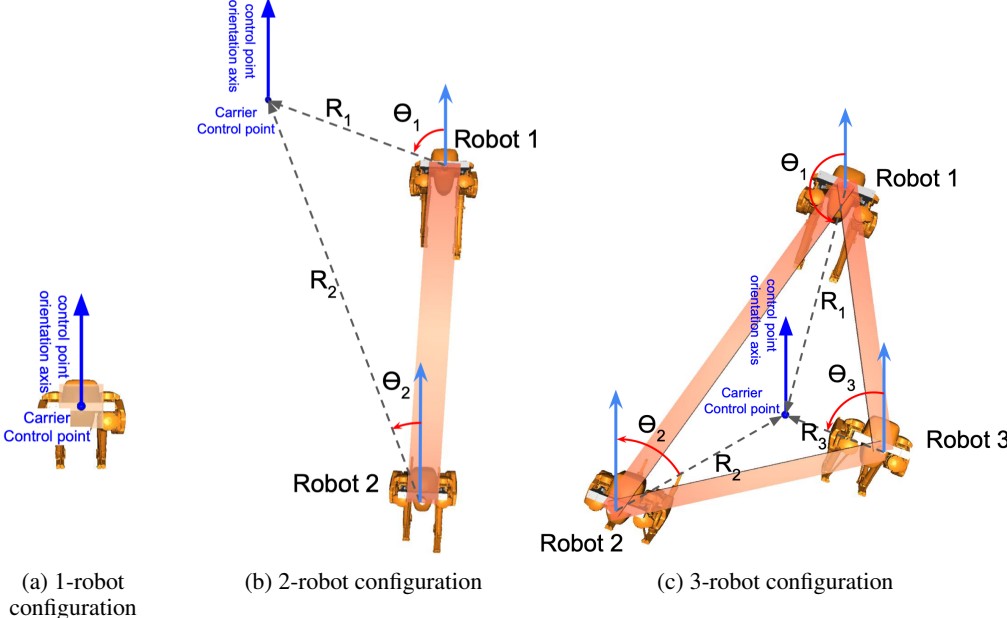

| (a) 1-robot configuration | (b) 2-robot configuration | (c) 3-robot configuration |

Figure 6: Training configuration using 1, 2 and 3 robots

| Parameter | Ranges |
|---|---|
| X velocity ($V_x$) | [-0.5, 2.0] $m/s$ |
| Y velocity ($V_y$) | [-0.3, 0.3] $m/s$ |
| Angular velocity ($\omega$) | [-$\pi/8$, $\pi/8$] $rad/s$ |
| Height ($h$) | [0.5, 0.8] $m$ |
| $R_i$ | [0, 3.5] $m$ |
| $\Theta_i$ | [-$\pi$, $\pi$] $rad$ |
| Commands Duration | [100, 450] timesteps |
| Force Duration | [50, 200] timesteps |
| Rigid bar masses | [0, 10] $kg$ (1-robot configuration) |
|  | [0, 20] $kg$ (2-robot configuration) |
|  | [0, 15] $kg$ (3-robot configuration) |

Table 5: Parameters ranges for training

# B Reward details

Below, we outline the reward terms used in our implementation. The reward components are divided into two categories: Local rewards $r_i^L$ are designed to produce stable locomotion for each robot, described in the local robot frame. The Global rewards $r^G$ are designed for following carrier commands with the combined action of all robots in the control point frame.

| Category | Reward Term | Definition | Weighting |
|---|---|---|---|
| $r_i^L$ 

 Local - Robot 

 $i \in \{1, \ldots, N\}$ | Feet airtime | $\begin{cases} 0.0 & \text{if } c_t = 0 \\ \sum_{f \in (l,r)} \|t_{f,air}^i - 0.35\| \cdot \mathbb{I}_{td,f}^i & \text{else} \end{cases}$ | 1.0 |
| | Feet contact | $\begin{cases} (n_c^i == 2) + 0.5 * (n_c^i == 1) & \text{if } c_t = 0 \\ (n_c^i == 1) & \text{else} \end{cases}$ | 0.1 |
| | Feet stance X | $\begin{cases} exp(-10 * \|feet_{l,x}^i - feet_{r,x}^i\|) & \text{if } c_t = 0 \\ exp(0) & \text{else} \end{cases}$ | 0.02 |
| | Feet stance Y | $\begin{cases} exp(-5 * \|feet_{l,y}^i - feet_{r,y}^i - 0.385\|) & \text{if } c_t = 0 \\ exp(0) & \text{else} \end{cases}$ | 0.02 |
| | Feet orientation | $exp(-30 * \sum_{f \in (l,r)} q_d(feet_{f,rpy}^i, base_{rpy}^i))$ | 0.15 |
| | Relative yaw orientation | $-\|\theta_t^i\|/\pi$ | 0.5 |
| | Joint force | $\begin{cases} exp(-0.2 * \|\|joint_{force,xy}^i\|\|) & \text{if } c_t = 0 \text{ and } N > 1 \\ exp(0) & \text{else} \end{cases}$ | 0.1 |
| | Base Height | $-\|base_z^i - h\|$ | 0.2 |
| | Base acceleration | $exp(-0.01 * \sum \|base_{accel,xyz}^i\|)$ | 0.1 |
| | Action difference | $exp(-8 * \sum \|a_t^i - a_{t-1}^i\|)$ | 0.1 |
| | Torque | $exp(-\sum \|\tau^i\|/\tau_{max}^i)$ | 0.05 |
| $r^G$ 

 Global - Carrier | X,Y velocity | $exp(-2 * (\hat{V}_{xy} - V_{xy}))$ | 0.15, 0.1 |
| | Orientation | $-q_d(O_{c,rpy}, \omega), \; exp(-30 * q_d(O_{c,rpy}, \omega))$ | 2.0, 0.15 |

Table 6: Description of reward terms

Notations used in Table 6 for the reward description.

- $N$ = Number of robots in the system
- $c_t$ = Carrier command consisting of linear velocity, angular velocity, and base height. $c_t = (V_x, V_y, \omega, h)$
- $c_t = 0$ means a hold still command
- $t_{f,air}^i$ = air time for a particular feet
- $\mathbb{I}_{td,f}^i$ = boolean variable indicating a touchdown for a particular feet
- $n_c^i$ = number of feet in contact with the ground
- $base^i$ = Robot pelvis global pose
- $feet^i$ = Pose of each foot in robot base frame
- $q_d(\cdot)$ = Quaternion distance function
- $\theta_t^i$ = Yaw orientation of each robot relative to the Carrier
- $joint_{force}^i$ = Force sensor reading on the ball joint connection
- $a_t^i$ = Action taken by the policy
- $\tau^i$ = Motor torques

- $O_c$ = Control point on the carrier
- $\hat{V}_{xy}$ = Current linear velocities of the control point

## C  Termination condition

An episode is terminated if any of the following conditions are met:

- The carrier's pitch or roll $O_{c,rp}$ is outside the range $[-30°, 30°]$.
- The robot's pelvis pitch or roll $base_{rp}^i$ is outside the range $[-30°, 30°]$.
- The robot's knee collides with the ground or with each other.
- The relative yaw orientation $\theta_t^i$ of each robot is outside the range $[-30°, 30°]$.
- The robot's pelvis height $base_z^i$ is outside the range $[0.5, 1.0]$
- The episode duration exceeds 500 timesteps (10 seconds).

## D  Dynamics Randomization

To facilitate transfer from simulation to the real robot, we employ dynamics randomization throughout all stages of the curriculum during training. This involves randomizing various physical parameters of the simulated environment, such as joint damping, link masses, center of mass positions, encoder noise, ground friction, terrain variations, and adding noise to states and policy rate. Table 7 lists the dynamics randomization ranges used.

| Element | Range |
|---|---|
| Joint Damping | $[-50, 250]\ \%$ |
| Body Mass | $[-25, 25]\ \%$ |
| CoM Position | $[-1, 1]\ \%$ |
| Friction | $[-20, 20]\ \%$ |
| Encoder Noise | $[-0.05, 0.05]\ rad$ |
| Ground Slope | $[-0.05, 0.05]\ rad$ |

Table 7: Dynamics randomization ranges

## E  PPO Hyperparameters

Table 8 lists the hyperparameters used for the PPO training.

| Hyperparameter | Value |
|---|---|
| Learning Rate (Actor & Critic) | 3e-4 |
| PPO Clip Range | 0.2 |
| Number of Epochs | 5 |
| Batch Size (Episodes) | 32 |
| Discount Factor (Gamma) | 0.95 |
| GAE Lambda | 1.0 |
| Value Function Coefficient | 0.5 |
| Entropy Coefficient | 0.01 |
| Max Grad Norm | 0.05 |
| Buffer Size | 60000 |

Table 8: PPO Hyperparameters

