# OpenReview forum: "Learning Decentralized Multi-Biped Control for Payload Transport"
_robot-learning.org/CoRL/2024/Conference — CoRL 2024_

### Official Review · Reviewer_eduL · 2024-07-21

**Originality:** 3
**Technical Quality:** 4
**Clarity Of Presentation:** 4
**Potential Impact:** 3
**Recommendation:** 4
**Confidence:** 4

**Review:**

# Strengths
* The paper is well-written and easy to follow.
* The method demonstrates robust policy when deployed on real robots.
* The framework has potential to work with unstructured and challenging terrains.
* The nature of decentralized control make it possible to be applied to varying number robots that are attached to a carrier in arbitrary stable configurations. The scalability is validated by sufficient experiments.
* The training curriculum design is unique and interesting for mutli-robot tasks.
* Extensive experiments with various configurations and varying numbers of robots are provided in simulation.
* The comparisons between decentralized multi-robot policy to centralized policy is valuable.

# Weaknesses
* The paper does not provide comparisons with previous control-based methods for multi-robot payload transport. Adding discussion or baselines in the experiments would be beneficial.
* Training is performed and validated with no more than 3 robots.
* The presented framework relies on fixed and stable configurations to carry on payloads.
* The tested experiments are on the flat ground, not including more challenging terrains.
* The authors directly apply the existing RL framework (IPPO) without major revisions.
* No quantitative and baseline results are provided for real robots.

**Quality Of The Limitations Section:**

3

**Questions For Rebuttal:**

* It looks like the the robots are distributed with symmetry under payloads as the number of agents increases in Figure 2. Would the robustness of the policy be affected when this symmetry is disturbed?
* Another potential benefit of applying decentralized multi-agent policy to robotic scenarios is better sim-to-real transfer, as indicated in the following work. It would be beneficial to discuss this in the results or consider it in the future work.
    * Dynamic Handover: Throw and Catch with Bimanual Hands. Huang et al., CoRL 2023.

**Robotics Focus:**

4

**Summary Of Paper:**

This paper designs a decentralized controller for multi-biped robot carriers, which can be applied to varying number of robots and various configurations. The controller is trained through a decentralized multi-agent reinforcement learning method, independent PPO (IPPO). The proposed method are evaluated on extensive settings in simulation, and achieve good performance with regards to drift, failure rate, power consumption for payload transport, reaching flexible and robust policy compared to baselines. The proposed framework is also validated on real robots.

**Summary Of Recommendation:**

This paper presents a solid work on multi-biped payload transport with multi-agent reinforcement learning. Extensive experiments are implemented to show its effectiveness in both simulation and on real robots.

---

### Official Review · Reviewer_wdBL · 2024-08-01

**Originality:** 3
**Technical Quality:** 4
**Clarity Of Presentation:** 4
**Potential Impact:** 3
**Recommendation:** 3
**Confidence:** 5

**Review:**

Strengths:
- The sim-to-real deployment constrains the multi-agent controller to making practical design choices for deployment, which was not the case in prior works
- The decentralized architecture and training scheme allow the approach to scale to varied numbers of robots and robot configurations without retraining
- Evaluation shows the expected trends of decreasing failure rate, decreasing energy consumption, and increasing payload capacity with number of robots
- Limitations are well addressed
- Overall, evaluations are complete and thorough, and the paper is well-written and novel

Weaknesses:
- It would be worth more discussion on the real-world practical applications of the approach. What kinds of application tasks and scenarios would it be desirable for multiple bipeds to collaboratively manipulate a payload? Why is this better for practical applications than just having e.g. a big quadruped? Since this paper only demonstrates the capability on flat ground, it might be hard for the reader to envision how such systems can evolve in the future, which would potentially make the paper more engaging. There would also be many practical drawbacks to a collaborative bipedal system such as high cost, many points of failure (which is not completely resolved by decentralization, since a biped that has a mechanical failure or runs out of battery still needs to be carried by the others), difficulty of perception with many surrounding robots (which makes decentralization less appealing)

**Quality Of The Limitations Section:**

2

**Questions For Rebuttal:**

The comparison with centralized and specialized policies seems to mostly suggest that the learning dynamics of the decentralized policy are better than those of the centralized policy or specialist policies. But from a capacity standpoint, it seems like the centralized and specialist policies should have strictly higher potential than the decentralized one, because they have access to more information about the system and task respectively. I wonder what would happen if you pretrained the policies using the decMBC approach before continuing training with a centralized or specialized architecture. Currently the results suggest that decentralization is actually better and it would be great to attribute the cause. Can you discuss the results of section 5.2 a bit and whether I am thinking about this reasonably?

Can you add more discussion of the practical applications as mentioned above?

**Robotics Focus:**

4

**Summary Of Paper:**

This paper proposes a decentralized controller for a variable number of bipedal robots to collaboratively manipulate payloads, with sim-to-real transfer.

**Summary Of Recommendation:**

The paper is well written, thorough, and novel, with high relevance to robotics. It would be enhanced by additional discussion of the results and the practical applications of its problem setting.

---

### Official Review · Reviewer_vVS6 · 2024-08-01
**Interesting paper**

**Originality:** 3
**Technical Quality:** 3
**Clarity Of Presentation:** 4
**Potential Impact:** 3
**Recommendation:** 3
**Confidence:** 3

**Review:**

Advantages:

1. This paper introduces the first decentralized controller designed for scalable bipedal robots. The experimental results, both in simulation and with real robots, robustly support the effectiveness of the proposed framework.

2. The paper presents thorough experimental results, demonstrating the framework's capability to
   a. control multiple robots and generalize effectively to out-of-distribution (OOD) numbers of robots.
   b. The trained policy adapts well to different payload shapes.
   c. Superior performance is evident in both simulated and real-world scenarios.

3. The paper is well-written, with a logical and fluent flow of ideas.

4. The authors clearly and honestly present the limitations of their work, potentially inspiring further research.

Disadvantages:

1. The paper's experiments are confined to flat surfaces. Given the inherent flexibility of bipedal robots, the motivation and experimental scope are somewhat misaligned. Testing on diverse terrain types would better align with the stated advantages of bipedal robots.

2. The framework employs a multi-stage training process, yet the paper lacks detailed ablation studies to elucidate the significance of each training stage. This omission raises concerns about reproducibility. Including such studies would help readers understand how each training method impacts the model's performance and generalization to different shapes and robot numbers.

3. The paper compares the proposed method with only a limited number of baselines. Given the numerous multi-robot control methods available, it would be beneficial for the authors to include more baselines for a comprehensive comparison.

**Quality Of The Limitations Section:**

3

**Questions For Rebuttal:**

Please refer to the above section.

**Robotics Focus:**

4

**Summary Of Paper:**

This paper presents a decentralized controller designed for varying numbers and configurations of rigidly attached bipedal robots, specifically within a payload transport scenario, without necessitating retraining. The authors provide simulation-based quantitative metrics to demonstrate the effectiveness of their approach across a diverse array of transport scenarios. Furthermore, the controller's capabilities are validated in real-world experiments involving systems composed of two and three Cassie robots. Notably, this work represents the first instance of a scalable multi-biped payload transport system.

**Summary Of Recommendation:**

This paper presents the first decentralized controller designed for varying numbers and configurations of rigidly attached bipedal robots, specifically within a payload transport scenario, without necessitating retraining. The authors provide simulation-based quantitative metrics to demonstrate the effectiveness of their approach across a diverse array of transport scenarios. Furthermore, the controller's capabilities are validated in real-world experiments involving systems composed of two and three Cassie robots. Notably, this work represents the first instance of a scalable multi-biped payload transport system. However, the paper suffers from the absence of ablation study/ insufficient baselines.

---

### Decision · Program_Chairs · 2024-09-04

**Decision:**

Accept

**Comment:**

Strengths:
- First scalable decentralized controller for multi-biped payload transport, showing effectiveness in both simulation and real-world experiments
- Robust generalization to out-of-distribution robot numbers and payload shapes without retraining
- Well-written paper with thorough experimental results and honest discussion of limitations
- Novel training curriculum design for multi-robot tasks

Weaknesses:
- Experiments limited to flat terrain, misaligned with motivation of bipedal robots for rough terrain
- Lack of detailed ablation studies on multi-stage training process impacts reproducibility
- Limited comparison to other multi-robot control method baselines
- Insufficient discussion of practical real-world applications and advantages over alternatives like large quadrupeds